# GUIDED SKETCH-BASED PROGRAM INDUCTION BY SEARCH GRADIENTS

## ABSTRACT

Many tasks can be easily solved using machine learning techniques. However, some tasks cannot readily be solved using statistical models, requiring a symbolic approach instead. Program induction is one of the ways that such tasks can be solved by means of capturing an interpretable and generalizable algorithm through training. However, contemporary approaches to program induction are not sophisticated enough to readily be applied to various types of tasks as they tend to be formulated as a single, all-encompassing model, usually parameterized by neural networks. In an attempt to make program induction a viable solution for many scenarios, we propose a framework for learning parameterized programs via search gradients using evolution strategies. This formulation departs from traditional program induction as it allows for the programmer to impart task-specific code to the program 'sketch', while also enjoying the benefits of accelerated learning through end-to-end gradient-based optimization.

## 1 BACKGROUND

For many tasks, machine learning can faithfully be applied with great results as the data usually has some form of statistical structure to facilitate it. These tasks are usually tabular in nature, although there are impressive demonstrations on datasets that are not so statistical, such as image classification and generation. Such general usage is in part due to deep neural networks whose underlying mathematical model essentially makes them universal function approximators. However, there are many cases where DNNs cannot be readily applied, ranging from computational and data requirements (George et al., 2017), generalizability (Lake et al., 2015), and even general lack of necessity in favor of simpler methods.

In fact, it may actually be necessary to utilize a non-neural alternative due to certain requirements such as interpretability and stability, things which are inherent in handwritten programs. Of course, in some cases it may be difficult to write certain programs from hand, making it necessary to automatically construct them from data. Two approaches that deal with the automatic construction of programs given a requirement, or specification, are program induction and program synthesis.

Program induction and program synthesis are two closely related techniques that aim to automate the process of generating programs from high-level specifications or examples. These techniques address the challenge of creating software without manually writing every line of code, thereby increasing efficiency and reducing human errors in the programming process. Both program induction and program synthesis have significant applications in various domains, including software development (Solar-Lezama, 2008), artificial intelligence (Silver et al., 2019), robotics (Lázaro-Gredilla et al., 2019), and more.

### 1.1 PROGRAM INDUCTION

Program induction involves the task of automatically constructing a *single* program from input-output examples or other forms of guidance. It is inspired by the idea of mimicking human learning, where people can generalize from specific examples to generate more general rules or concepts. In the context of program induction, a system attempts to learn a program that can produce the desired outputs for a given set of inputs.

One of the key challenges in program induction is finding a balance between overfitting (producing programs that work only for the provided examples) and underfitting (producing overly general programs that fail on unseen inputs), thus program induction generally requires some degree of domain knowledge with respect to the problem.

Various techniques have been applied to the problem of program induction, including probabilistic programming (Lake et al., 2015) and neural controllers for computers (Graves et al., 2014; Łukasz Kaiser & Sutskever, 2016; Graves et al., 2016).

## 1.2 PROGRAM SYNTHESIS

Program synthesis takes a broader approach by aiming to generate programs from high-level specifications, which can include natural language descriptions, formal logic expressions, or other forms of input that describe the desired behavior of the program. Unlike program induction, in which the system learns a program for a specific task, program synthesis is concerned with generating correct programs of arbitrary, high-level specifications.

Program synthesis can be categorized into two main types: deductive synthesis and inductive synthesis (Gulwani et al., 2017).

### 1.2.1 DEDUCTIVE SYNTHESIS

In deductive synthesis, the system constructs a program by applying formal reasoning based on logical rules and constraints. It involves techniques from formal verification and automated theorem proving in order to progressively deduce a program that satisfies the given problem. The process often starts with a high-level formal specification (although not strictly necessary) and involves step-by-step transformations to deduce a concrete program. Deductive synthesis is not a different type of program synthesis, but a technique of synthesizing programs. Thus, one can leverage deductive synthesis techniques in an inductive synthesis framework.

### 1.2.2 INDUCTIVE SYNTHESIS

Inductive synthesis is the process of constructing programs through generalization from examples. The examples are typically hard input-output pairs, but they can also be more relaxed, such as through the use of a loss function. This approach is particularly useful when a high-level formal specification alone is not sufficient, and some examples or demonstrations are needed to guide the synthesis process. Most contemporary program synthesis solutions are inductive due to the greater flexibility that comes with this approach. Moreover, the idea of programming-by-example overlaps with machine learning, allowing for data-driven synthesis of programs using standard machine learning models.

## 2 CONTEMPORARY CHALLENGES IN PROGRAM INDUCTION AND PROGRAM SYNTHESIS

While program induction and program synthesis offer promising avenues for automating program generation, they come with their own set of challenges. These challenges can often hinder their widespread adoption in practical applications.

### 2.1 SEARCH SPACE COMPLEXITY

One of the primary challenges in program induction and synthesis is the sheer complexity of the search space for potential programs. In both cases, the space of possible programs can become astronomically large, making it computationally infeasible to explore all possibilities exhaustively. To address this issue, domain-specific languages (DSLs) are often introduced to constrain the search space, allowing more efficient exploration.

## 2.2 VALID PROGRAM GENERATION

Generating valid programs from a vast search space is a non-trivial task. Constructing grammars for DSLs and ensuring that all programs adhere to these grammars can be complex. The difficulty increases when designing DSLs for specific problem domains, as they often require custom interpreters or compilers. Nevertheless, explicit grammar representation can significantly increase the quality of the output when used in combination with various program learning methods (Bunel et al., 2018).

## 2.3 PROGRAM REPRESENTATION

The choice of how to represent programs plays a crucial role in program induction and synthesis. Finding a suitable representation that balances expressiveness and efficiency is essential. Some representations, such as neural networks (Devlin et al., 2017b; Balog et al., 2017; Ellis et al., 2020), can be highly flexible but may introduce additional complexity and overhead. Fortunately, this issue can be mitigated when the system is formulated with the right representation Devlin et al. (2017a).

## 2.4 OPTIMIZATION CHALLENGES

Efficiently optimizing potential candidate solutions in the program search space is another challenge. The variable-length nature of programs can make optimization non-trivial. Some solutions involve using recurrent neural networks to suggest candidate solutions, but this adds another layer of complexity, which may not be desirable for tasks that require rapid prototyping. Currently, program searching is limited to enumeration methods for unguided synthesis / induction, while guided synthesis methods make use of neural networks. In an ideal world, the program search would be guided using mathematically sound optimization algorithms like gradient descent.

## 3 PROGRAM INDUCTION VIA EVOLUTION STRATEGIES PT.1

In light of these challenges, this work introduces a novel framework for program induction, which leverages the concept of "sketching" (Solar-Lezama, 2008). This approach allows a programmer to provide an abstract codebase, or "sketch", that contains "holes" that represent incomplete code sections. These "holes" can be updated through the search algorithm to satisfy the specification.

Sketching offers several advantages in program induction:

### 3.1 BENEFITS OF SKETCHING

- **Reduced Search Space**: Sketching narrows down the search space for program induction algorithms. By providing a high-level structure or partial code, the algorithm is guided towards more relevant solutions, making the search for a valid program more efficient.

- **Error Reduction**: Sketching helps in reducing the chances of generating incorrect or undesirable code. By providing a partial program, the algorithm is steered towards producing a more reliable and well-structured solution.

- **Customization**: Sketching allows for customizing the generated program to suit specific needs or preferences. It allows fine-tuning of the algorithm's output based on requirements and preferences.

- **Data Efficiency and Generalization**: A well-designed sketch can lead to more generalizable solutions. Instead of producing a specific program for a single instance, the algorithm may generate programs that work for a broader range of inputs or scenarios. This was demonstrated in a previous work (Lake et al., 2015), where a probabilistic program of handwritten characters Lake et al. (2019) was learned by fitting probability distributions on drawing data. The subsequent model was then able to generate and recognize models from as little as one example, as well as generate completely new concepts.

This approach to program induction is especially appealing due to its very close resemblance to machine learning: the holes are parameters of the model, which is represented by the program

sketch. As it follows, the searching algorithm is the optimizer that updates the sketch so that the resulting program best satisfies the specification, which is the objective function.

By interpreting the program induction problem through the lens of statistical optimization, it becomes much easier to come up with an optimization scheme that works effectively with programmatic functions.

Computer programs have a very general formulation, so standard differentiable optimization schemes do not work out of the box. Instead, black-box methods are employed in order to account for the variable nature of programs, which may or may not be mathematically grounded. The particular black-box optimizer that is used is Natural Evolution Strategies Wierstra et al. (2011), which accelerates parameter search by computing of search gradients. This is great since it makes it possible to use standard gradient descent optimizers to efficiently search through potentially countless programs.

## 4 PROGRAM INDUCTION VIA EVOLUTION STRATEGIES PT.2

A program is composed of a set of parameters $\phi = (\phi_0, \phi_1, \phi_2, ..., \phi_N)$ that indicate tokens, such as a method or function parameter. Each $\phi_n$ may take a variety of values with respect to the program structure. For example, a function involving transformations on images will take values in the range of 0-255 for each of the red, blue, green and alpha channels.

Thus, the process of finding an optimal program is reduced to solving for the optimal parameters, which can become very difficult as complexity of the program sketch increases.

For example, a moderately sized program with 10 discrete tokens, each taking 4 possible values, will require enumerating over $4^{10} = 1,048,576$ programs, which is too large to efficiently evaluate through brute-force search. When the tokens are of continuous nature, searching no longer becomes tractable.

One way to search for programs is through rejection sampling from the program writer. However, this approach does not guarantee optimal solutions, leading us to look towards smarter optimization algorithms.

The problem of program *synthesis* can be viewed as a minimization of the following objective:

$$\min_{\theta} \mathbb{E}_{spec \sim P(spec)} [D_{KL}(P_{GT}(prog|spec)||P_{\theta}(prog|spec))] \tag{1}$$

That is, by minimizing the KL divergence, the parameterized program converges to the ground-truth program.

The problem is that we cannot directly minimize the above divergence simply due to the fact that we do not have access to the ground-truth distribution to begin with (i.e. we do not have the correct programs).

To that end, we can maximize an approximation of the above objective in a variational manner:

$$\max_{\theta} \mathbb{E}_{(prog,spec) \sim P(prog,spec)} [\log(P_{\theta}(prog|spec)))] \tag{2}$$

Considering that the program writer $P_{\theta}(prog|spec)$ is parameterized (e.g. by a recurrent neural network), one can simply optimize the parameters in a supervised fashion such that they satisfy program-specification pairs. However, the program *induction* problem traditionally does not have a parameterized program writer nor does it have access to program-specification pairs. The only piece of data that is given is the program specification, thus the above objective needs to be re-framed to account for this fact:

$$\max_{\theta} \mathbb{E}_{(in,out) \sim P(spec)} [\log(P_{\theta}(out|in))] \tag{3}$$

Equation (3) is very similar to Equation (2) in that optimization of the objective is a matter of training the parameterized model on the specification through supervised learning. The only difference is that the program writer *is* the program itself, removing the need to sample a dataset from the writer.

To ensure converging behaviour, one can make use of gradient descent optimization. Traditional derivative-based optimization cannot be used though as it assumes that the underlying program is differentiable, which may or may not be the case. To ensure that the program can be optimized regardless of its underlying structure, black-box optimization is needed.

Evolution strategies, particularly Natural Evolution Strategies (NES) (Wierstra et al., 2011) are a class of black-box optimization methods that perform gradient descent by approximating the gradient (or in the case of NES, the *natural* gradient). The approximation is a result of sampling perturbed parameters from a *search distribution*, evaluating said parameters and then computing the gradient as an expectation over the scored search gradients. As the search distribution is what gets updated, the model being sampled from the search distribution can take any arbitrary structure so long as the tokens are sampled from a differentiable probability distribution (e.g. categorical distribution).

The gradient estimation process can be described using the following process:

$$\nabla_\theta J(\theta) \approx \frac{1}{\lambda} \sum_{k=1}^{\lambda} f(x_k) \nabla_\theta \log(\pi(x|\theta))$$

$$\theta \leftarrow \theta + \eta \nabla_\theta J(\theta)$$

Where $\nabla_\theta J(\theta)$ is the gradient of the objective function with respect to search distribution parameters $\theta$, $f(x_k)$ is the objective function, and $\nabla_\theta \log(\pi(x|\theta))$ is the score of the search distribution with respect to $\theta$. The NES gradient estimator is in essence the REINFORCE estimator (Sutton et al., 1999) applied in the parameter space as opposed to the model's sample space. The search distribution parameters can be updated using a standard gradient descent optimizer like Adam (Kingma & Ba, 2017).

In the context of program induction, we specify a search distribution over the token space of the various tokens that make up the program. Learning amounts to sampling a variety of programs, evaluating each program with respect to the specification, and then updating the search distributions so that they progressively sample better programs.

The reader will notice that this process resembles program synthesis, where the search distribution is a program writer that is updated iteratively to converge to the ground-truth distribution. The difference is that we are maximizing Equation (3) as we do not have access to the full joint probability, and the writer samples programs actively instead of passively maximizing likelihood.

Choosing a search distribution for the tokens is rather simple since tokens are usually of categorical nature for discrete values, and normally distributed for continuous values. For the normal distribution, the closed-form solution from Salimans et al. (2017) is used. For the categorical distribution, we derive the score function as the gradient of the softmax function.

Because of their general nature and applicability towards many types of data, these two distributions are sufficient to model a complex variety of programs, provided that the program sketch is rich enough to capture certain behaviours. Even in cases where the true parameter distribution is different, the simple parameterization and theoretical guarantees (e.g. central limit theorem) ensure a well enough approximation. We demonstrate the practicality of evolution strategies through a set of experiments incorporating mixed discrete and continuous holes in the program sketch.

---

**Algorithm 1** Categorical Natural Evolution Strategies

---

1: **Input:** Learning rate $\alpha$, initial search distribution parameters (logits) $\theta_0$
2: **for** $t = 0, 1, 2...$ **do**
3:     Initialize gradients $T = ()$
4:     Sample $\epsilon_1, ...\epsilon_n \sim \pi(\epsilon|\theta_t)$
5:     Compute returns $F_i = F(\epsilon_i)$ for $i = 1, ..., n$
6:     Compute probabilities $P_i = \pi(\epsilon_i|\theta_t)$ for $i = 1, ..., n$
7:     **for** $i = 0, ..., n$ **do**
8:         **for** $j = 0, ..., |\theta|$ **do**
9:             **if** $\epsilon_i = j$ **then**
10:                 $L_i = P_i * (1 - P_i) * F_i$
11:             **else**
12:                 $L_i = -P_i * \pi(j|\theta_t) * F_i$
13:             **end if**
14:             $T_i = T_i + \frac{1}{n}L_i$
15:         **end for**
16:     **end for**
17:     $\theta_{t+1} \leftarrow \theta_t + \alpha T_i$
18: **end for**

---

## 5 EXPERIMENTS

### 5.1 SIMPLE PROGRAM INDUCTION

To demonstrate the feasibility of the evolution strategies approach to program induction, the first experiment involves learning a simple if-else program with continuous holes for numerical values and discrete holes for mathematical operators (greater-than, addition, division etc.). As simple as this problem may be, it is rather difficult to solve using more traditional approaches simply due to the presence of continuous variables. To that end, this first experiment serves as the perfect benchmark to verify that program searching with search gradients is a viable method for program induction.

To generate the specification and for verification purposes, the following Rust program is used:

```rust
fn ground_truth_prog(x: f32) -> f32
{
    if x > 3.5
    {
        return 4.2 * x;
    }

    return x * 2.1;
}
```

A specification of four input-output pairs were created using the input set [1.0, 2.0, 4.0, 5.0].

To build the sketch, we introduce the holes for the conditionals and the return statements as such:

```rust
fn synth_prog(x: f32) -> f32
{
    if x [COND] [Real]
    {
        return [Real] [OP] x;
    }

    return x [OP] [Real];
}
```

Where [COND] indicates one of three conditionals (equal, greater-than and less-than), [OP] indicates one of four mathematical operators (addition, subtraction, multiplication and division), and

[Real] corresponds to any real number. To model these holes, we use the categorical distribution for the [COND] and [OP] fields, and the normal distribution for the [Real] fields. This amounts to six different search distributions that need to be optimized. Setting aside the real number fields, there conditional and operation fields result in $3 * 4^2 = 48$ possible programs.

Training-wise, the mean-squared error (MSE) loss is used to compare the ground-truth outputs to the outputs of the generated programs. The scores are then standardized to have a Gaussian distribution as it is necessary for optimal performance. The SGD optimizer is used with a learning rate of 0.1 for 10,000 iterations.

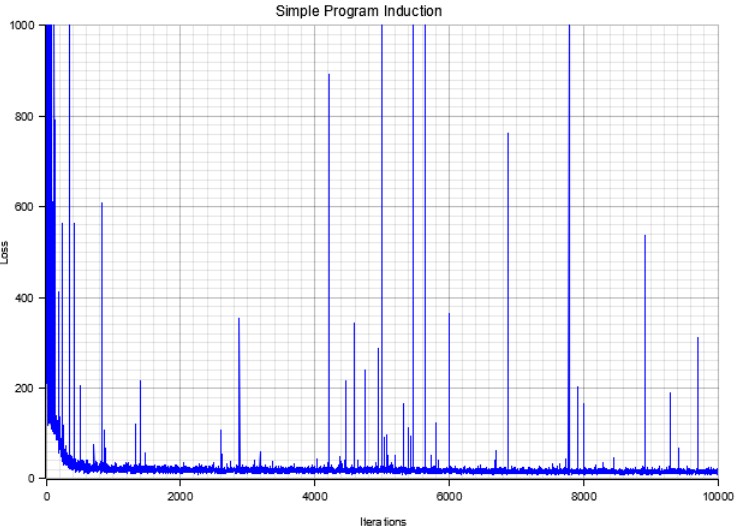

Figure 1: Loss of parameterized program over 10,000 iterations. Lower loss indicates better program behaviour.

The learned program can be visualized by taking the argmax of the search distributions. In this case, the program learned from the sketch is the following:

```
fn synth_prog(x: f32) -> f32
{
    if x < 2.2305248
    {
        return 2.4594104 * x;
    }

    return x * 4.0324993;
}
```

Although there is some discrepancy between the ground-truth program and the learned program with regards to the continuous variables, the two programs are equivalent for all intents and purposes. In particular, the system has learned to create an equivalent program with the "less-than" conditional, swapping the return statements of the ground-truth program to account for this fact.

## 5.2    PROGRAM INDUCTION WITH MULTIPLE INPUTS

The second experiment increases the complexity of the program induction task, as it requires the search distributions to optimize a specification involving multiple inputs. Like the first experiment, a ground-truth program is created to generate the specification as well as to serve as a reference to compare to the induced program.

The same procedure as the first experiment is used, where the loss function used is mean squared error and score standardization is employed to accelerate learning. The SGD optimizer is used with a learning rate of 0.0995, and the number of iterations is extended to 20,000 steps.

The ground truth program constructed is as follows:

```
1  fn ground_truth_prog(x1: f32, x2: f32) -> f32
2  {
3      if x1 > x2
4      {
5          return 2.0 * x1 + x2;
6      }
7
8      return 2.0 / x2 - x1;
9  }
```

And the corresponding sketch that is trained:

```
1  fn ground_truth_prog(x1: f32, x2: f32) -> f32
2  {
3      if x1 [COND] x2
4      {
5          return [Real] [OP] x1 [OP] x2;
6      }
7
8      return [Real] [OP] x2 [OP] x1;
9  }
```

The specification contains the set of two inputs [(5.8, 2.5), (5.0, 6.2), (7.4, 6.1), (5.5, 9.4)], resulting in a set of outputs [14.1, -4.677419, 20.9, -5.287234].

The training graph of the experiment is drastically different from the first experiment however, showing instability across the training steps as indicated by the spikes in training loss.

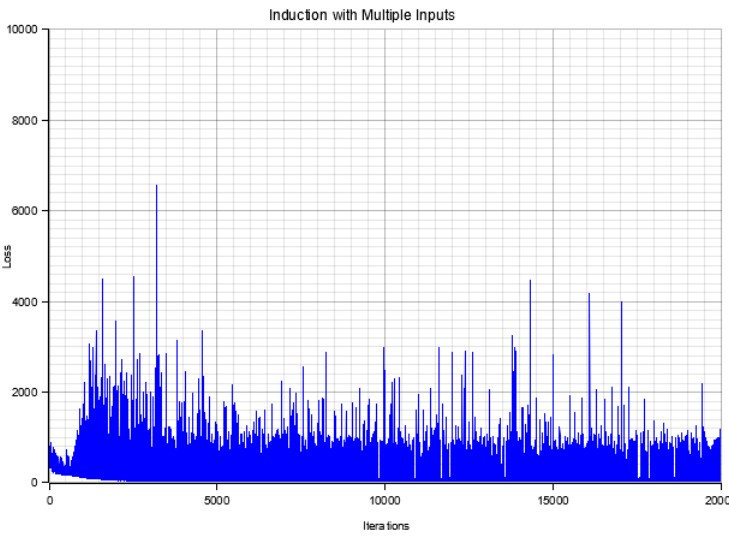

Figure 2: Loss of parameterized program over 20,000 iterations. Lower loss indicates better program behaviour.

This is reflected in the argmax program, which has learned much different operators and constants, demonstrating overfitting behaviour. Nevertheless, the outputs of the induced program approaches the ground-truth outputs, highlighting the optimality of the solution for the given specification.

```
1  fn synth_prog(x1: f32, x2: f32) -> f32
2  {
3      if x1 < x2
4      {
5          return 14.287576 / x1 - x2;
6      }
7
8      return 8.472884 * x2 / x1;
9  }
```

## 6 CONCLUSION AND DISCUSSION

In this paper, a novel method for program induction using search gradients is proposed. This method allows programmers to construct a sketch of a program that is optimized in a mathematically sound way. This approach facilitates the use of standard gradient descent optimizers, bringing the benefits of machine learning into program induction while being simple and easy to implement.

With all that said, there are several drawbacks with this method. For one, the programs explored in this work are trivial as they make use of basic operations. While more complex programs involving loops were attempted, they were unable to train due to infinite looping. This absence of formal verification is one of the major reasons for the inability to construct sufficiently complex programs.

Another limitation is the inability to handle dynamically sized programs. Due to the nature of evolution strategies, it is necessary to have a fixed program size in order to store gradients for program updates. Currently, the behaviourally similar genetic algorithms are able to handle this property through crossover, which evolution strategies lack. One future avenue is to explore combining genetic algorithms with evolution strategies in order to bring the best of both worlds.

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
