# OpenReview forum: "Guided Sketch-Based Program Induction by Search Gradients"
_ICLR.cc/2024/Conference — Submitted to ICLR 2024_

### Official Review · Reviewer_EcME · 2023-10-30

**Soundness:** 2 fair
**Presentation:** 2 fair
**Contribution:** 1 poor
**Rating:** 1
**Confidence:** 5

**Summary:**

This paper proposes using Natural Evolution Strategies for synthesizing programs. Given a program sketch, they create a parameterized distribution for possible values of the holes. They attempt to use Natural Evolution Strategies to learn, via gradient descent, a distribution whose argmax will be the ground truth program. On two example synthesis problems, the authors show their approach leads to solutions which satisfy the input/output specifications.

**Strengths:**

I think there could be potential in the approach of using Natural Evolution Strategies for program synthesis.

The authors have a good grasp of related work and the key difficulties behind face program synthesis and program induction.

**Weaknesses:**

Overall, the paper is still in its infancy.

The evaluation consists of running the approach on two toy synthesis tasks, each of which consists of a single conditional. They are more like example problems rather than a benchmark from which the performance of the approach can be gleaned.

There is no comparison to alternative methods.

The approach is described with little detail. In particular, the core contribution is using Natural Evolution Strategies (NES) to fill program sketches, and the details and intuition of how NES works  here is not systematically explained, so it's hard to understand what is going on.

I would suggest evaluating your approach on some standard benchmarks and comparing to prior work, so we can get a sense of how well NES works in terms of computational cost and performance.

**Questions:**

None

---

### Official Review · Reviewer_MY2o · 2023-10-30

**Soundness:** 2 fair
**Presentation:** 2 fair
**Contribution:** 2 fair
**Rating:** 3
**Confidence:** 3

**Summary:**

The paper proposes an approach to automated program synthesis based on
user-provided sketches that allow to learn parameterized programs. The authors
describe their approach, which uses evolutionary algorithms and gradients, and
evaluate it empirically.

**Strengths:**

The paper tackles an interesting problem, but falls short in several areas.

**Weaknesses:**

First, the description of the proposed method is unclear. Algorithm 1 contains
several concepts, e.g. the return F_i, that are not defined or explained in the
text. It is impossible to even judge, never mind reproduce what the authors have
done without this information.

Further, the empirical evaluation is tiny, comprising only two programs unless I
missed something. This isn't sufficient to show the general applicability of the
method, in particular as there is no comparison to other approaches. It is
impossible to judge if or how much the proposed approach improves over the
literature and other approaches.

It is unclear why the description of the method was broken into "PT.1" and
"PT.2".

**Questions:**

No questions.

---

### Official Review · Reviewer_y6cE · 2023-11-03

**Soundness:** 1 poor
**Presentation:** 1 poor
**Contribution:** 2 fair
**Rating:** 3
**Confidence:** 4

**Summary:**

This work proposes an approach to program induction based on reducing program induction to an optimization problem using Natural Evolution Strategies, which is an approach based around approximating gradients through perturbing and scoring parameters to update a search distribution from which programs can be sampled. Program tokens are sampled from categorical distributions while floating point constants are sampled from normal distributions, and the parameters of these distributions are optimized through the search process. This search method is used to fill the holes in a program sketch, where the size of the output program is known.

**Strengths:**

* The idea of a program synthesis approach based around gradient descent on a search distribution which is then used for sampling programs is generally an interesting one

**Weaknesses:**

**Overall** This work is very preliminary, the experiments are very limited and much of the paper is spent on background around synthesis approaches. I don't think this is ready for publication at ICLR, but I encourage the authors to submit with revisions to a workshop.

Specific points:
- The experiments are very preliminary – they run two experiments, each involving finding a single program. More complete experiments would be needed on larger datasets (ideally ones used in the synthesis community) to properly evaluate the method
- Working with fixed sized programs is a considerable limitation, for example a hole can be filled with something like `5` or `+` or but not with `(+ 2 x)`. It's difficult for me to think of a setting where this kind of thing is known.

Some more minor points around the background and setup of the paper:
- The framing of "program induction" as being distinct from "program synthesis" doesn't align with how I believe the terms are generally used in the field – program induction is a specific type of program synthesis. The authors give a reasonable definition of program induction in section 1.1, however in section 1.2 they say "Unlike program induction, in which the system learns a program for a specific task, program synthesis is concerned with generating correct programs of arbitrary, high-level specifications.". I would instead frame it as program synthesis being the general setting of finding a program that solves a task given a specification of that task, and program induction being program synthesis when the specification is underspecified (eg when it's given as input-output examples, as opposed to a formal logical spec). I'm happy to provide references to synthesis papers that use the terms in these ways if it's helpful, I'm fairly confident these are the usual definitions and that synthesis is not generally held in contrast to induction but rather as a supertype of it. Also, note that program synthesis still generally deals with solving tasks – the high level specifications are just another way of specifying tasks.
    - I also notice "inductive synthesis" is described in section 1.2.2 which I think is describing the same thing as program induction again – Are these two sections meant to be describing different ideas?
- Deductive synthesis is described in 1.2.1 then never comes up again and doesnt seem related to the papers approach
- Generally, much of sections 1-3 can be condensed and combined into fewer headings.
- In section 1 there's a mention of how "interpretability" is "inherent in handwritten programs". This statement could use a bit more of a caveat – it's certainly possible (and in fact common) to write programs that are difficult to understand/interpret, depending on what you mean by interpretability. For example, obfuscated code and uncommented code can be difficult for humans to understand, and the amount of static analysis / automated interpretability one can do of programs is highly dependent on the language. I agree with the argument that programs are often much more interpretable than deep learning models, but I would suggest caveating it a bit more than just stating that interpretability is "inherent"

**Questions:**

* Can you clarify the program induction / synthesis distinction mentioned in Weaknesses above?

---

### Official Review · Reviewer_a6L3 · 2023-11-09

**Soundness:** 3 good
**Presentation:** 2 fair
**Contribution:** 2 fair
**Rating:** 3
**Confidence:** 4

**Summary:**

This work introduces a method for performing program induction by using a blackbox optimisation for
finding the best parameters to fill holes in parametrised sketches of programs.

**Strengths:**

The paper is very clearly written and task it's trying to solve well-specified. While there is
plenty of work on program synthesis with sketches, I do feel there are still many powerful applications
of sketching that have been under-explored like this work.

**Weaknesses:**

The use of a blackbox optimization algorithm was novel, but the programs involved are so
small. Would it not be possible to make a set of differentiable program sketches by enumerating
all the discrete choices, running a gradient optimization for each of them, and then taking
the sketch which optimised the objective function best? For programs this small, isn't an
enumerative search still viable?

I also wish the proposed algorithm had been compared against something else. A simple
baseline or maybe even RobustFill? Right now, the experiments consist of two fairly small
programs and no sense of whether the results shown are better or worse than other methods
used for program induction.

**Questions:**

Would it be possible to cover this algorithm to neurally-guided methods?
Would it be possible to cover it to some enumerative search baseline?
Could these comparisons include number of programs explored and wall-clock time for the search procedure?

---

### Meta-Review · Area_Chair_CM8R · 2023-12-06

**Metareview:**

The paper proposes using Natural Evolution Strategies, a search and optimization technique, to find useful completions to program sketches. The paper does not seem to provide sufficient details about the proposed approach. The experiments are extremely limited in scope, and do not provide comparisons against existing methods for this type of problem. The authors did not respond to any of the concerns raised by the reviewers.

**Justification For Why Not Higher Score:**

The reviewers raised multiple concerns about the paper, and the authors did not engage in a discussion to address any of those concerns.

**Justification For Why Not Lower Score:**

N/A

---

### Decision · Program_Chairs · 2024-01-16

Reject